# The Impact of Borrowing Size on the Economic Development of Small and Medium-Sized Cities in China

**Xiaoxia Gong [1] and Fanglei Zhong [2],***

[1]  School of Management, Northwest Minzu University, Lanzhou 730030, China; 273972033@xbmu.edu.cn
[2]  School of Economics, Lanzhou University, Lanzhou 730000, China
*  Correspondence: zfl@lzu.edu.cn; Tel.: +86-139-9312-7625

**Abstract:** Sharing the economic agglomeration effects of large cities is considered an effective way to enhance economic growth in small and medium-sized cities, yet there remains a lack of relevant quantitative empirical research. In this study, the three dimensions of borrowing size of 285 prefecture-level cities and nighttime lighting data from 2004 to 2013 in China are used to assess its effect on economic development by a fixed-effects model with panel data. The results show that first, the effect of borrowing size on small and medium-sized cities' economic development is significantly positive. In contrast, the effect of borrowing size on large cities is not significant. Second, the magnitude of the impact of borrowing size on small and medium-sized cities varies considerably across regions. In middle and western China, the most significant positive effect is from borrowing economic activity density and borrowing advanced functions. However, in eastern China, the most significant positive effect is from borrowing population, while borrowing advanced functions has a significant negative effect. In the northeast, borrowing economic activity density has a significant positive effect. This article provides policy recommendations in three areas: improving intercity accessibility, rationalizing the layout of urban industries, and supporting the development of education.

**Keywords:** borrowing size; small and medium-sized cities; city network; prefecture-level city; regional coordination

## 1. Introduction

The development of small and medium-sized cities is an important aspect of healthy urban agglomeration in metropolitan areas [1]. As a key factor connecting urban and rural areas—the development of small and medium-sized cities—is related to the specific implementation of the new urbanization program in China [2]. Its development is not only an important starting point for balancing efficiency and fairness in relation to high-quality urban development, but also an important means of promoting coordinated regional development [3,4]. Thus, the "Key Tasks for New Urbanization Construction in 2019" issued by the National Development and Reform Commission of China on 31 March 2019 stated that to achieve high-quality urbanization, attention must be paid to the development of various-sized cities in urban agglomerations, and thus the development of small and medium-sized cities should be supported. The "Opinions of the Central Committee of the Communist Party of China and the State Council on Establishing and Improving the Urban-Rural Integration Development System, Mechanism and Policy System" issued on 15 April 2019 proposed that urban agglomeration should be the main form of promotion of the coordinated development of large, medium-sized, and small cities, and small towns.

However, the development of small and medium-sized cities is not consistent across the various regions in China. For example, small cities in the Yangtze River Delta and the Pearl River Delta dominate the top 100 counties in the country [4–6], while "the poverty belt around Beijing and Tianjin" and "the lamp shadow area" continue to exist [3,7]. Traditional theories have mainly analyzed the dual roles of large cities in the agglomeration of and diffusion toward neighboring small cities, emphasizing the dominant position of large

cities in the urban system, while ignoring the active role that small and medium-sized cities can play in the urban system [8–10]. How do we improve the economic growth of small and medium-sized cities? What are the reasons for the differences in the levels of economic development of small and medium-sized cities? In this study, we introduce the concept of "borrowing size" and analyze the impact of borrowing size on the economic development of small and medium-sized cities from the perspective of the national urban network [11]. Using nighttime lighting data for 285 prefecture-level cities in China from 2004 to 2013, we use empirical analysis to identify both favorable and unfavorable factors related to the economic development of small and medium-sized cities in an effort to identify shortcomings and provide policy recommendations that will help to promote the healthy economic development of these cities, thereby helping to coordinate China's new urbanization and regional development and increase both efficiency and equity.

## 2. Literature Review and Hypotheses

There has long been debate on whether China's urbanization program should prioritize the development of large cities or that of small and medium-sized cities. Scholars advocating the development of large cities have conducted their analyses from the perspective of agglomeration and economies of scale [8,9,12], stating that the expansion of the city scale can increase the benefits provided by agglomeration [13–17]. However, the development of big cities also creates diseconomies of scale such as diminishing marginal returns from factor inputs [18] and urban environmental pollution [19,20]. Scholars who advocate the development of small and medium-sized cities believe that the priority development of these cities is conducive to rapid urbanization [21], which can reduce the psychological costs of rural labor entering the city and achieve a better combination of urban and rural markets [21,22].

Numerous scholars have focused on the issue of urban scale and economic development over a long period of time. China's urbanization program has experienced problems related to insufficient growth and structural imbalance, with the latter embodied in the insufficient or even shrinking scale of small and medium-sized cities [15]. Shanzi Ke conducted an empirical study using panel data from prefecture-level cities and found that cities need to reach a certain threshold scale to obtain the benefits of industrial linkages [12]. However, most prefecture-level cities in China have not yet reached the threshold [15,23]. It is difficult for small and medium-sized cities to obtain scale and agglomeration benefits [10,24,25]. However, some studies have shown that the size of a city does not seem to limit its development. For example, a large number of small and medium-sized cities in China's Yangtze River Delta urban agglomeration have an average GDP growth rate that is higher than the regional average growth rate and that of some large cities [5]. This finding does not conform to the traditional theory of agglomeration economics. Meijers et al. [1,26,27] found that agglomeration has a positive effect on productivity. However, other factors unrelated to city size are also important in promoting the benefits of urbanization.

Given the increasing level of connection among cities, studies on urban networks have become increasingly popular [6,28–30]. In 1950, Zonneveld first proposed the concept of an "urban network" based on the economies of scale produced by improved transportation and information channels [30–32]. In 1974, Rolfs proposed that the phenomenon that the product value increases as the number of its users increases should be termed "network externality" [33]. "Urban network externality" involves the expansion and extension of the concept of network externality used in the field of urban economics [34–36]. Capello [37] was the first to propose the concept of urban network externality, stating that cities can obtain economic benefits through the synergy and complementarity generated by the functional networks constructed among cities. Urban network externalities can be either positive or negative. Positive urban network externalities are mainly reflected in "borrowing size" [38]. In 1973, the famous planner and economist Alonso put forward the concept of "borrowing size" in "Urban Zero Population Growth" [39]. From the perspective of the urban system, he believed that we should examine not only the optimal size of a single



city, but also the scale of cities that interact with each other [40–42]. Hepworth found that the development of communications technology had enhanced the borrowing capacity of small cities around London, which had led to the deurbanization of London [43]. Phelps examined the relationship between the local economy of small towns and neighboring large cities from the perspective of borrowing size and found that increased borrowing size enabled enterprises to achieve transformation and development in the context of rural urbanization [40,44].

Subsequent studies found that borrowing size in small cities is not entirely dependent on their proximity to large cities [38,45]. Proximity to large cities does not necessarily enable small cities to obtain development opportunities through increased borrowing size [44,46]. Some scholars have divided the borrowing size of small towns into three types. The first type is small towns adjacent to big cities, who can borrow size from the big cities; the second is a cluster of adjacent small towns, who can borrow size from one another; and the third is small towns that are neither adjacent to big cities nor adjacent to other small towns, who can borrow size through the city network. The principle of borrowing size provides various development strategies for different types of small towns [47]. Yao and Song used China's eight national-level urban agglomerations and surrounding non-urban prefecture-level cities as research objects, and using systematic generalized method of moments and other empirical estimation methods found that small and medium-sized cities can enjoy the benefits of a large-city-generating agglomeration economy by borrowing size [5]. However, urban network externalities are not always beneficial to city development [10,44,46]. Meijers et al. pointed out that the negative effects of urban network externalities manifest as agglomeration shadows [7,47–50] that mainly arise from competition among cities [51–55]. Small cities face pressure from large cities, as well as pressure from neighboring small cities in a competition for survival, which limits their development [52]. Bindong Sun and Song Ding used 108 small cities in the Yangtze River Delta urban agglomeration in China as research objects and found no direct evidence for the existence of agglomeration shadows, providing empirical support for the concept of borrowing size [53]. Therefore, urban network externalities suggest that borrowing size and agglomeration shadows can exist simultaneously. However, for a given city, the intensities of these two effects generally differ [10,48].

Our review of the literature revealed the following findings. First, the research objects of most previous studies on borrowing size are developed countries in Europe and America in the latter stages of urbanization, with few studies focused on developing countries, even though numerous developing countries are currently undergoing a period of rapid urbanization, especially China. Second, there are disagreements regarding the concept of borrowing size. Scholars have tended to define the concept based on their own research requirements, rather than applying a universally accepted definition of borrowing size, which has created a degree of confusion. Third, at present, the research objects in studies on borrowing size in China are mainly at the national level of urban agglomerations. Few scholars have analyzed borrowing size from the perspective of the four major regions (the east, middle, west, and northeast) in China or the respective levels of economic development in northern and southern China. Fourth, in terms of setting distance weights, Chinese scholars currently use one of two methods: The first involves using Euclidean distances, while the second involves evaluating the intercity network connections based on train travel time within the network. Both methods have some limitations.

Following Yao and Song [5], in this study we define borrowing size as the process whereby a given city accesses scale benefits through participating in a network with other cities or regions. Borrowing size includes three dimensions: borrowing population, borrowing economic activity density, and borrowing advanced functions. Adopting the perspectives of spatial interaction theory, absolute advantage theory, comparative advantage theory, flow space theory, and network externality theory, we analyze the mechanism by which borrowing size influences the economic development of small and medium-sized cities and propose and test several hypotheses.

Borrowing population is related to potential market size. The larger the borrowed population, the larger the markets for both production factors and products. This provides an important source of economies of scale in small and medium-sized cities, and helps to offset the shortcomings of these cities and promote their economic development [56]. For producers in small and medium-sized cities, the larger the borrowed population, the larger and more elastic the labor market, which increases the probability of matching people with jobs. An abundant labor supply can also reduce employment costs to some extent [57], thereby reducing production costs and enhancing competitiveness. For businesses in small and medium-sized cities, the size of the borrowed population represents the size of the potential consumer market. The larger the borrowed population, the larger the potential market for the sale of goods and services. For consumers, the expansion of the potential market is conducive to finding jobs that match their skills, thereby increasing income and raising consumption levels. However, the borrowed population is a relatively static indicator that does not necessarily reflect the level of economic activity, and thus it has an insignificant effect on the economic development of small and medium-sized cities.

**Hypothesis 1:** *An increase in population can promote the economic development of small and medium-sized cities, but the effect is relatively insignificant.*

Economic activity density refers to the knowledge spillovers between various industries and reflects the degree of urbanization of the economy (the Jacobs externality effect [58]). As knowledge is public and non-exclusive, when companies use their knowledge to carry out innovative activities and increase productivity, they do not reduce the effectiveness of that knowledge. Further, companies in other industries can also benefit from this process. However, knowledge spillovers are, to some extent, constrained by space and the effects of localization [59]. Therefore, only by strengthening the links between economic activities in all walks of life and enabling enterprises to move closer to knowledge-intensive areas can advanced knowledge be transformed into production efficiency. However, if firms move closer to knowledge-intensive areas, on the one hand, the cost of acquiring knowledge decreases, but on the other hand, rental costs increase, thereby offsetting the benefits of acquiring new knowledge. Therefore, companies distant from knowledge-intensive areas (generally major cities) rely on urban networks to strengthen their connections with those cities, that is, increase their borrowing of economic activity density to acquire knowledge, promote innovation, and improve production efficiency [60,61].

**Hypothesis 2:** *An increase in economic activity density can promote the economic development of small and medium-sized cities.*

The economic relationships between cities are mainly facilitated by urban functions [59]. The impact of borrowing advanced functions on the development of small and medium-sized cities is mainly achieved through the division of labor among functional spaces in those cities. From the perspective of the industrial chain, major cities and small and medium-sized cities have strengthened their economic ties through the division of labor, increasing their economies of scale and enabling large, medium-sized, and small cities to form symbiotic and mutually beneficial relationships. Entrepreneurs in the intercity network can share facilities such as warehousing and business services to improve efficiency and optimize resource allocation. Consumers in small and medium-sized cities can access entertainment, culture, education, and other functions in the central city through network connections, which satisfies the diversity of consumer demand while enabling consumers to continue to live in small and medium-sized cities where housing costs are lower, thereby maximizing their personal utility. However, if a region has a preference for a specific high-level industry, cities in that region will gradually deviate from their previous local comparative advantages because of the homogeneous industrial structure. Intense competition between industries and market segmentation will reduce the overall

economies of scale in the region [62], which will reduce the original innovation incentive provided by the attraction of advanced industries and limit improvements in urban productivity to a certain extent.

**Hypothesis 3:** *Borrowing advanced functions has a dual effect on the economic development of small and medium-sized cities.*

## 3. Data and Methods

### 3.1. Data Sources and Processing

Nighttime lighting intensity provides an objective measure of daily economic activity, and includes some informal areas of the economy that are not included in GDP estimates. Therefore, in this study, we use nighttime lighting density to measure the level of urban economic development [63]. The nighttime lighting data were obtained from 34 reports issued over the 22-year period from 1992 to 2013 by the National Geophysical Data Center within the National Oceanic and Atmospheric Administration. The nighttime lighting used in this study mainly includes lights from cities, towns, and other locations with continuous lighting (including gas flares), while short-term lighting events such as fires, sun glare, and the aurora were removed. The background noise is set to zero, and the range of DN (Digital Number) is 1–63. However, the original nighttime lighting data cannot be directly used to study cities. Thus, we adopted the method proposed by Zhang and Pan [64] to implement continuity correction, pixel saturation, and abnormal value processing in relation to the nighttime lighting data from 2004 to 2013. The grid values of the nighttime lighting data are summed using the prefecture-level city as the boundary, and the total DN value of the nighttime lighting of the prefecture-level city in the study area is obtained, followed by the per capita DN value. The processed nighttime lighting data better reflects the economic development of the prefecture-level city, and is more appropriate for cross-year and cross-regional comparisons.

The economic data were mainly obtained from the 2005–2014 editions of the China City Statistical Yearbook. Due to incomplete data for some prefecture-level cities (such as Haidong, Tongren, Bijie, Lhasa, and Sansha), we eliminated those regions, leaving a sample of 285 prefecture-level cities. Although there have been some administrative name changes in some prefecture-level cities, we have retained the original names to maintain the integrity of the sample.

The scope of this study includes prefecture-level cities, rather than municipal districts. Regarding the weighted distance matrix, we do not use a spatial distance matrix, but rather use a 285 × 285 matrix composed of the shortest highway routes between cities. The data used to determine these routes include 81,225 distance values obtained from 2345 Intercity Highway Distance Search Tool (http://tools.2345.com/jiaotong/lc.htm). We used MATLAB software to measure the three dimensions of borrowing size, and used ArcGIS to display and analyze the temporal and spatial distributions of borrowing size and nighttime lighting data.

### 3.2. Measurement of Borrowing Size

On the basis of previous domestic and foreign studies, we used the cities in the national urban network to measure the three dimensions of borrowing size (borrowing population, borrowing economic activity density, and borrowing advanced functions) as follows:

borrowing population (bpop):

$$\text{bpop}_i = \sum_{m=1}^{284} \frac{\text{pop}_m}{w_{i,m}}, \forall i \neq m \tag{1}$$

borrowing economic activity density (bden):

$$bden_i = \sum_{m=1}^{284} \frac{den_m}{w_{i,m}}, \forall i \neq m \tag{2}$$

borrowing advanced functions (bfun):

$$bfun_i = \sum_{m=1}^{284} \frac{fun_m}{w_{i,m}}, \forall i \neq m \tag{3}$$

where i and m represent two cities in the national city network, $pop_m$ represents the population of city m, $den_m$ represents the economic activity density of city m, $fun_m$ represents the advanced functions of city m, and $w_{i,m}$ represents the shortest distance by road between city i and city m.

To measure economic activity density (den), we draw on the method proposed by Yao and Song [5] and use the density of urban passenger transport systems. Thus, economic activity density = (total annual urban railway, highway, water, and air passenger transport)/built-up areas/365.

Advanced functions (fun) are usually measured by the proportion of the total labor force in a city that is engaged in high-end occupations. On the basis of data availability and Liu's research [3,42], we used high-end service industries (including transportation, storage and delivery, information transmission, computing services and software, finance, real estate, leasing and business services, scientific research, technical services, and geological prospecting industries) and the ratio of the number of employees in these industries to that in the manufacturing industry to measure borrowing advanced functions.

### 3.3. Measurement Model Settings

This study mainly explores the impact of the three dimensions of borrowing size (borrowing population, borrowing economic activity density, and borrowing advanced functions) on the economic development of small and medium-sized cities. The benchmark model is set as

$$lightp_{i,t} = \beta_0 + \beta_1 bpop_{i,t-1} + \beta_2 bfun_{i,t-1} + \beta_3 bden_{i,t-1} + \beta_4 lnpop_{i,t-1} + \beta_5 lnpop2_{i,t-1} + \beta_6 pinvest_{i,t-1} + \beta_7 pfd_{i,t-1} + \beta_8 pedu_{i,t-1} + \varepsilon_{i,t} \tag{4}$$

where t represents the year and i represents the i-th prefecture-level city. The explained variable lightp represents urban productivity, while the independent variable bpop is the population that can be borrowed by the prefecture-level city, bfun is the advanced functions that can be borrowed by the prefecture-level city, bden is the economic activity density that can be borrowed by the prefecture-level city, lnpop is the scale of the prefecture-level city, lnpop2 is the quadratic term of lnpop, pinvest is the fixed asset investment level, pfdi is the foreign direct investment level, and pedu is the local human capital level.

It must be pointed out that given the possibility of mutual causality between the prefecture-level city's current borrowing size, population, advanced functions, economic activity density, capital investment, foreign direct investment, human capital and other variables, and current urban productivity, the independent variables in this model are all processed with a lag. To reduce the heteroscedasticity of the model, the population variables in relation to prefecture-level cities take a logarithmic form.

### 3.4. Index Selection and Data Description

Regarding the explained variables, we drew on Liu Xiuyan's approach [3,42] and used per capita nighttime lighting brightness (lightp) to indicate the level of urban economic development, making the results more objective and complete.

Regarding the explanatory variables, the three dimensions of borrowing size (bfun, bpop, and bden) are all calculated using the method described in Section 3.2.

Regarding the control variables, on the basis of the literature review, we used the logarithmic values of the populations of prefecture-level cities (lnpop), while fun is the

local advanced functions, den is the local economic activity density, and the fixed asset investment level (pinvest) is the amount of investment in fixed assets included in the current year's GDP. pfdi is the proportion of foreign direct investment in the current year's GDP in CNY, pedu is the proportion of college students among the total population of the prefecture-level city, and pgov is the proportion of government expenditure included in the year's GDP.

Regarding the comparability and completeness of the data, the data used in this study mainly come from the China City Statistical Yearbook (2005–2014). Given the possibility of an inverted U-shaped relationship between urban population and urban productivity, the quadratic form of the urban population logarithm (lnpop2) was used.

## 4. Results

### 4.1. Spatiotemporal Evolution of Nighttime Lighting Density and Borrowing Size

Figure 1 shows the spatiotemporal distribution of nighttime lighting density in the prefecture-level cities studied in 2004, 2009, and 2013. In the figure, the areas with higher levels of nighttime lighting brightness per capita are mainly distributed in economically developed areas or areas with an average level of economic development and relatively small populations. For example, in 2004, the only prefecture-level cities with a per capita nighttime lighting brightness DN value of more than 20 units were Shenzhen, Dongguan, Jia Yuguan, and Karamay, with the first two cities located in economically developed regions and the other two in regions with relatively small populations. By 2009, Zhongshan City, Foshan City, Shenzhen City, Wuhai City, and Ordos City were included in the list of cities with a per capita nighttime lighting brightness DN value exceeding 20 units. By 2013, the number of prefecture-level cities whose per capita nighttime lighting brightness DN value exceeded 20 units had increased to 15. From the spatial perspective, cities with high levels of per capita nighttime lighting brightness tend to be distributed in coastal areas, the middle and lower reaches of the Yangtze River area, and the Bohai Bay area, which is basically consistent with the economic development levels of China's small and medium-sized cities.

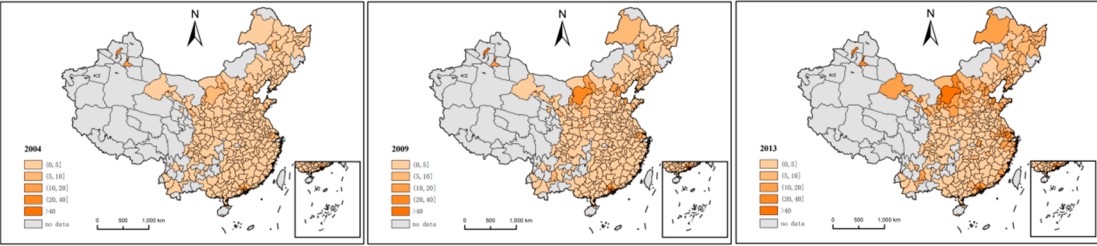

**Figure 1.** Spatiotemporal distribution of nighttime lighting density.

Figure 2 shows the spatiotemporal distribution of borrowing population by the prefecture-level cities in the study area. It can be seen that cities borrowing population are mainly concentrated in the North China Plain and the Middle and Lower Yangtze Plains. These areas were developed earlier and are now densely populated, including the most populated provinces such as Henan, Hebei, and Shandong. In addition, these regions have relatively flat terrain, dense road networks, and good regional accessibility. Therefore, prefecture-level cities in these regions borrow relatively large populations. Cities that borrow populations of less than 50 units are mainly distributed in the northwestern frontier areas, such as Urumqi and Karamay. In terms of temporal distribution, the level of borrowing population has remained relatively stable overall, with only a slight increase. This stability in relation to the spatiotemporal distribution of borrowing population mainly reflects the slow growth rate of the Chinese population as a result of the decline in the birth rate in recent decades.

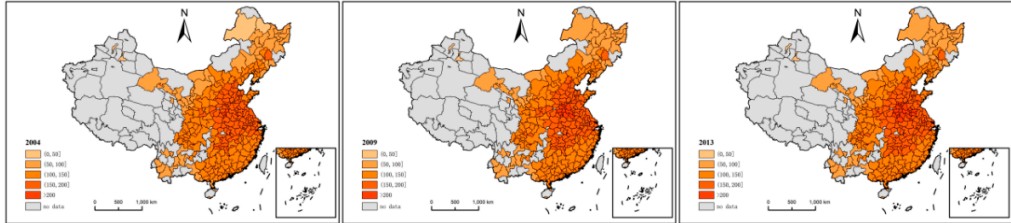

**Figure 2.** Spatiotemporal distribution of borrowing population.

Figure 3 shows the spatiotemporal distribution of borrowing advanced functions by prefecture-level cities in the study area in 2004, 2009, and 2013. It can be seen that the number of prefecture-level cities with higher levels of borrowing advanced functions has declined, then increased, and then declined again. This has mainly been caused by the relative strength of the high-end service industry and the manufacturing industry. It is well-known that the global financial crisis triggered by the subprime mortgage crisis in the United States in 2007 had a dramatic impact on industrial development worldwide. The crisis caused a substantial reduction in the orders of China's exporters, which in turn led to business closures, bankruptcies, and mass layoffs. When facing a financial crisis, the general manufacturing industry is more vulnerable than the high-tech industry [65]. Therefore, in 2009, when the advanced functions of the prefecture-level cities were larger, the advanced functions that could be borrowed were also larger. By 2013, following the recovery of the global economy and China's emphasis on the development of the real economy, even though the high-end service industry had recovered to a certain extent, it remained relatively weak compared with the manufacturing industry. Thus, borrowing advanced functions by prefecture-level cities remained lower in 2013 than in 2009.

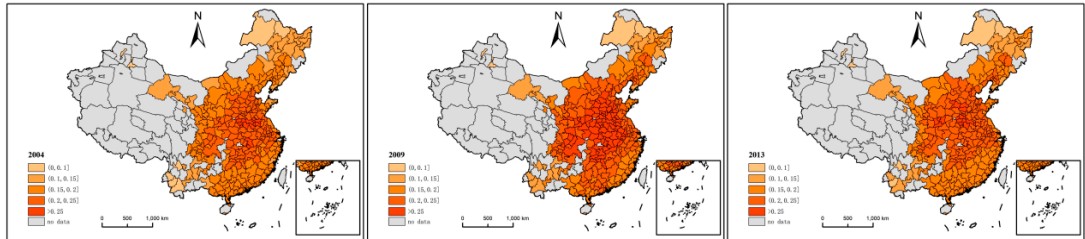

**Figure 3.** Spatiotemporal distribution of borrowing advanced functions.

Figure 4 shows the spatiotemporal distribution of borrowing economic activity density by prefecture-level cities in the study area. It can be seen that the number of prefecture-level cities with higher levels of borrowing economic activity density gradually increased. In 2004, there were only three prefecture-level cities with a level of borrowing economic activity density higher than 0.14 units, but by 2009 this number had increased to 12, and by 2013 it had increased to 46. In general, borrowing economic activity density is highly dependent on location-related factors. Areas with a higher level of borrowing economic activity density are located in the center of the studied area, while areas with a lower level of borrowing economic activity density are generally located at the edge of the studied area.

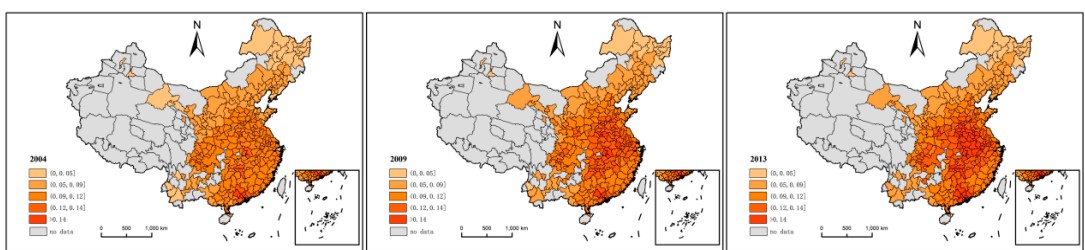

**Figure 4.** Spatiotemporal distribution of borrowing economic activity density.

*4.2. Regression Analysis*

To examine the impact of borrowing size on the economic development of small and medium-sized cities, among the 285 prefecture-level cities for which the data is available, we use the 254 small and medium-sized prefecture-level cities as the research objects while the other 31 large prefecture-level cities (including provincial administrative centers and municipalities directly under the central government in Shenzhen) were used as the research reference. Empirical testing based on the socioeconomic data and nighttime lighting data for prefecture-level cities in China from 2004 to 2013 was conducted using a fixed-effects model and panel data.

Table 1 shows the results of the regression analysis of the impact of borrowing size on urban economic development in cities at prefecture level and above (classified as small and medium-sized cities, and large cities, respectively). Overall, for small and medium-sized cities, the core explanatory variables are significant at the 1% level. Further, with the addition of core explanatory variables, the goodness of fit increases from 0.326 to 0.401, meaning that the explanatory power of the model has increased. From the perspective of the direction and effect of the core explanatory variables on the explained variables, borrowing population, borrowing economic activity density, and borrowing advanced functions all have a positive effect on the economic development of small and medium-sized cities, indicating that the economic development of these cities benefits from increased borrowing size. Small and medium-sized cities can promote economic development by borrowing population, economic activity density, and advanced functions contained in the national city network.

**Table 1.** Results of regression analysis using the fixed-effects model.

| | Medium and Small Cities | | | | Big City |
| --- | --- | --- | --- | --- | --- |
| | **fe1** | **fe2** | **fe3** | **fe4** | **fe5** |
| lbpop | 0.341 *** | | | 0.164 *** | 0.255 ** |
| | (0.0331) | | | (0.0445) | (0.1110) |
| lbfun | | 47.77 *** | | 19.93 *** | 4.845 |
| | | (7.5310) | | (6.7720) | (13.1000) |
| lbden | | | 70.50 *** | 41.17 *** | 28.39 |
| | | | (8.2100) | (7.5630) | (16.7400) |
| llnpop | 183.3 *** | 152.5 *** | 159.3 *** | 171.0 *** | 43.3900 |
| | (36.6000) | (37.9700) | (38.1000) | (37.5900) | (46.7100) |
| llnpop2 | −15.89 *** | −12.67 *** | −13.39 *** | −14.78 *** | −3.6400 |
| | (3.0850) | (3.2240) | (3.2050) | (3.1840) | (3.6180) |
| lfun | 0.1620 | 0.0411 | 0.138 * | 0.0972 | 1.997 ** |
| | (0.0987) | (0.0597) | (0.0828) | (0.0853) | (0.9650) |
| lden | 0.0188 | 0.0230 | 0.0646 *** | 0.0440 | −3.626 * |
| | (0.0358) | (0.0206) | (0.0206) | (0.0315) | (1.9540) |
| lpinvest | −1.395 *** | −1.251 ** | −0.962 ** | −1.755 *** | −2.510 *** |
| | (0.4980) | (0.4910) | (0.4540) | (0.5160) | (0.8720) |
| Lpfdi | −26.50 ** | −33.65 ** | −31.27 ** | −28.79 ** | −6.7320 |
| | (11.9300) | (13.0900) | (12.8100) | (11.9100) | (8.5070) |
| Lpedu | 3.3190 | 45.1000 | 35.8900 | 6.2160 | 26.1100 |
| | (25.3000) | (27.9300) | (26.6500) | (24.0000) | (16.5200) |
| Lpgov | 0.5690 | 1.4000 | 1.540 * | (1.3770) | 35.78 ** |
| | (0.7910) | (0.9580) | (0.8870) | (1.2470) | (14.3300) |
| Constant | −563.2 *** | −458.3 *** | −470.9 *** | −513.3 *** | −163.1 |
| | (110.3000) | (111.2000) | (112.3000) | (112.7000) | (152.2000) |
| Observations | 2286 | 2286 | 2286 | 2286 | 279 |
| R-squared | 0.371 | 0.326 | 0.363 | 0.401 | 0.652 |
| Number of city | 254 | 254 | 254 | 254 | 31 |

Notes: Robust standard errors are shown in parentheses; ***, **, and * represent $p < 0.01$, $p < 0.05$, and $p < 0.1$, respectively.

For large cities, only borrowing population has a significantly positive effect on economic development, while the effects of borrowing economic activity density and borrowing advanced functions on economic development do not pass the significance test. This means that the effects of borrowing economic activity density and borrowing advanced functions on urban economic development are mainly reflected in small and medium-sized cities. The main reason for this is that large cities are generally regional economic centers with a high degree of marketization and convenient transportation, which attracts the accumulation of capital, technology, and talent. To a certain extent, borrowing population can potentially provide human capital for the development of big cities. However, large cities have a high level of technology and are relatively advanced in terms of their service industry, which has a radiating and leading role in relation to small and medium-sized cities. Generally speaking, large cities are far apart, and the surrounding small and medium-sized cities are mostly focused on manufacturing industries. Therefore, large cities cannot be driven by small and medium-sized cities in terms of the development of technology and modern service industries.

Regarding control variables, for small and medium-sized cities the coefficients of the primary term lnpop and the quadratic term lnpop2 of the urban population logarithm are both significant at the 1% level. However, the coefficient for lnpop is positive and the coefficient for lnpop2 is negative, implying that from the perspective of the national urban network, there is an inverted U-shaped relationship between the local population of small and medium-sized cities and the level of urban economic development. In the initial stage of urban development, urban economic development benefits from the expansion of the local population, but in the later stages, urban economic development will be negatively affected by "big city malaise". This conclusion is consistent with the findings of numerous previous studies.

### 4.3. Spatial Heterogeneity

In this study, the cities included in the sample are divided into the four traditional major regions (northeast, east, middle, and west) and the southern and northern regions for heterogeneity testing to analyze the effects of regional differences in borrowing size on the economic development of small and medium-sized cities.

It must be pointed out that this study mainly draws on the results of Guo Aijun and Fan Qiao [66] when dividing the cities between the northern and southern regions. In this study, we divided the 285 prefecture-level cities into the northern and southern regions based on the geographic location of the provinces to which they belong. When the latitude of a provincial capital city was below 32° N, we classified the prefecture-level cities under the jurisdiction of its provinces as southern cities. In contrast, when the latitude of a provincial capital city was above 32° N, we classified the prefecture-level cities under the jurisdiction of its provinces as northern cities. These southern cities included 16 prefecture-level cities in Anhui Province, nine in Fujian Province, 21 in Guangdong province, 14 in Guangxi Province, four in Guizhou Province, two in Hainan Province, 12 in Hubei Province, 13 in Hunan Province, 13 in Jiangsu Province, 11 in Jiangxi Province, 18 in Sichuan Province, eight in Yunnan Province, and 11 in Zhejiang Province, as well as Shanghai City and Chongqing City. There was a total of 154 prefecture-level cities in the southern region. These northern cities included 12 prefecture-level cities in Gansu Province, 11 in Hebei Province, 17 in Henan Province, 12 in Heilongjiang Province, eight in Jilin Province, 14 in Liaoning Province, nine in Inner Mongolia Autonomous Region, five in Ningxia Hui Autonomous Region, one in Qinghai Province, 17 in Shandong Province, 11 in Shanxi Province, 10 in Shaanxi Province, and two in Xinjiang Uygur Autonomous Region, as well as Beijing City and Tianjin City. There was a total of 131 prefecture-level cities in the northern region.

Table 2 shows the influence of borrowing size on the level of economic development of small and medium-sized cities divided into the four traditional major sectors and the northern and southern regions. Overall, there are obvious differences in the impact of

borrowing size on the economic development of cities in the various sectors. As far as the central and western regions are concerned, the coefficient for borrowing advanced functions is significantly positive at the 5% level and the coefficient for borrowing economic activity density is significantly positive at the 1% level, indicating that the economic development of small and medium-sized cities is mainly the result of borrowing advanced functions and economic activity density. This means that the small and medium-sized cities in the central and western regions are relatively weak in terms of technological innovation capabilities. Further, the level of industrial development in these areas is low, and thus the upgrading of the industrial development needs to be accelerated. Through the city network, they can be connected to large cities with high-tech, scientific research, and cultural and other advanced industries, so that these small and medium-sized cities in conjunction with large cities provide a reasonable spatial layout in terms of industrial division of labor, which can strengthen the links between industries. As the economic activity density around small and medium-sized cities has increased, these cities have strengthened the links between industries through urban networks, which has increased the spatial spillover effect in terms of knowledge, technology, and information. Therefore, an increase in borrowing advanced functions and borrowing economic activity density is conducive to the formation of a shared pattern of development among small and medium-sized cities in the central and western regions and large cities. However, in the central and western regions, the coefficient for borrowing population is not significant. This is probably because the small and medium-sized cities in the middle and western regions are inadequate in terms of agglomeration capacity and incapable of attracting people, which means that the potential market size is unable to be effectively transformed into economic benefits.

**Table 2.** Urban economic development by borrowing size divided into the four traditional major plates and the southern and northern regions.

| | fe_w | fe_m | fe_e | fe_ne | fe_n | fe_s |
|---|---|---|---|---|---|---|
| Lbpop | 0.0552 | 0.0009 | 0.482 *** | 0.16 | 0.0557 | 0.273 *** |
| | (0.1390) | (0.0280) | (0.0870) | (0.0957) | (0.1010) | (0.0418) |
| Lbfun | 20.66 ** | 19.51 *** | −19.04 * | −1 | 29.58 *** | −9.947 *** |
| | (9.6600) | (5.9060) | (11.0900) | (4.8770) | (10.5000) | (2.8430) |
| Lbden | 43.78 *** | 42.33 *** | 9.66 | 117.9 *** | 50.72 *** | 7.371 |
| | (9.6700) | (6.0290) | (15.3900) | (25.6600) | (11.0500) | (6.1290) |
| Control-variable | YES | YES | YES | YES | YES | YES |
| Constant | −597.1 *** | −157.5 *** | −601.1 *** | −509.5 *** | −607.7 *** | −179.3 *** |
| | (137.3000) | (43.9300) | (217.0000) | (134.7000) | (130.4000) | (64.2400) |
| Observations | 657 | 666 | 684 | 279 | 1,044 | 1,242 |
| R-squared | 0.413 | 0.683 | 0.535 | 0.791 | 0.449 | 0.454 |
| Number of city | 73 | 74 | 76 | 31 | 116 | 138 |

Notes: Robust standard errors are shown in parentheses; ***, **, and * represent $p < 0.01$, $p < 0.05$, and $p < 0.1$, respectively. W, m, e, ne, n, and s represent the west, middle, east, northeast, north, and south, respectively.

For small and medium-sized cities in the eastern region, the coefficient for borrowing population is significantly positive at the 1% level and the coefficient for borrowing advanced functions is negative at the 10% level, reflecting the fact that the economic development of small and medium-sized cities in the eastern region is mainly affected by borrowing population and advanced functions. This is because the eastern region has a better natural environment, relatively flat terrain, a dense transportation network, and high population density. The eastern region opened up to the world earlier and has a high degree of marketization. Further, a large number of colleges and universities are congregated in the eastern region, and thus the population has advanced conceptual and technological innovation capabilities. All of these factors have enabled the rapid transformation of this huge potential market into economic advantages, which has increased the turnover of economic resources. In addition, although China has entered a stage of slow population growth, its population remains huge, providing a steady stream of labor for the economic

development of small and medium-sized cities in the east, thereby making the economy more sustainable. However, the eastern region, as a cluster of advanced functional industries, faces fierce competition in terms of advanced functions. Therefore, borrowing advanced functions will hinder the economic development of small and medium-sized cities to a certain extent as a result of this competition. As for borrowing economic activity density, the coefficient is not significant. This may be because the eastern region, as a high-tech industrial cluster, is affected by the homogenization of knowledge, technology, and information, making the spatial spillover effect less important for small and medium-sized cities in this region.

For small and medium-sized cities in the northeastern region, the coefficient for borrowing economic activity density is significant at the 1% level. Compared with the central and western regions, the coefficient is much larger, indicating that borrowing economic activity density plays a leading role in promoting the economic development of small and medium-sized cities in the northeastern region. Generally, areas with high economic activity density have frequent personnel exchanges, and technology and information can be exchanged and spilled more quickly. At the same time, areas with high economic activity density are also conducive to increasing industry specialization. Overall, the industrial structure in northeastern China is aging, with overcapacity in traditional industries and a low level of technology. This enables small and medium-sized cities with high economic activity density to quickly absorb advanced technology and information through the city network, increasing their ability to transform economic activity into economic benefits. What is more, the coefficient for the borrowing population is not significant, which may be because of the development environment in the northeast region. While China is in the process of adjusting its economic growth rates and industrial structure, the northeastern region, which is an important industrial base, is dominated by heavy industry, and the development of its light industry, private enterprises, and modern service industries is lagging. With the decline of heavy industry and the depletion of resources in the northeastern region, the attractiveness of this region to the population has been weakened, leading to significant urban contraction. Borrowing population to transform into an effective source of new product markets and human resources is difficult. Further, the coefficient for borrowing advanced functions is not significant. This may be because the traditional industries in the northeastern region, such as steel, coal, and building materials, are facing greater pressure in terms of overcapacity as a result of profound changes in market demand [67]. It is difficult for these industries to form effective industrial connections with other industries with advanced functions.

From the perspective of the northern and southern regions, the effect of borrowing size on the economic development of small and medium-sized cities differs. For small and medium-sized cities in the northern region, the coefficients for borrowing advanced functions and borrowing economic activity density are both significantly positive at the 1% level, while the coefficient for borrowing population is not significant. However, for small and medium-sized cities in the southern region, the coefficients for borrowing population and borrowing advanced functions are both significant at the 1% level. However, the coefficient for borrowing population is positive, while that for borrowing advanced functions is negative. In addition, the coefficient for borrowing economic activity density is not significant. The north–south differences in the impact of the three dimensions of borrowing size on the economic development of small and medium-sized cities reflect the differences in the economic development of northern and southern cities. Over the past 40 years since the reform and opening up, China has gradually been integrated into the Asia-Pacific marine economy. Southern cities, especially southeastern coastal cities, have developed rapidly, and the market system has been well cultivated. The collective changes in population, wealth, knowledge, skills, and consumption have been described as a "peacock flying southeast". The comprehensive strength of southern cities, with their excellent development opportunities and living environment, is constantly increasing. The improvement of technological innovation capabilities in southern cities has

promoted industrial transformation and upgrading, and improved quality and efficiency. Borrowing population provides southern cities with vast consumer markets and reserves of human capital. However, borrowing advanced functions has also generated fierce competition with foreign countries, causing some unnecessary losses, which to a certain extent has hindered the improvement of local productivity. In contrast, the economy of the northern region is dominated by resources, energy, and heavy industry, and thus the task of eliminating excess capacity is greater. Investment in research and development is relatively inadequate, the concept of development is backward, and economic development is relatively slow. Therefore, borrowing economic activity density is conducive to small and medium-sized cities absorbing advanced concepts, technologies, and information through urban networks. To a certain extent, borrowing advanced functions can avoid competition among homogeneous advanced functional industries, and thus enable small and medium-sized cities to achieve coordinated development through the spatial division of industries.

## 5. Conclusions and Policy Recommendations

### 5.1. Conclusions

On the basis of the results of previous studies, we constructed a city network composed of 285 prefecture-level cities in China and measured the three dimensions of borrowing size. Then, we analyzed changes in the spatiotemporal distribution of the three dimensions of borrowing size in 2004, 2009, and 2013. Using socioeconomic data and nighttime lighting data for China's prefecture-level cities from 2004 to 2013, we empirically tested several hypotheses using a fixed-effects model and panel data, and also compared the heterogeneity of the northeastern, eastern, central, western, southern, and northern regions to analyze the different regional effects of borrowing size on the economic development of small and medium-sized cities. The results enabled us to draw the following conclusions.

First, regarding the spatiotemporal distribution of borrowing size, areas with a higher level of borrowing population are more concentrated in space, and mainly located in the North China Plain and the Middle and Lower Yangtze Plain. In contrast, areas with a lower level of borrowing population are mainly located in the northwestern frontier region. Borrowing population has only slightly increased over time, with the overall situation remaining stable. The number of prefecture-level cities that borrowed advanced functions decreased, then increased, and then decreased again. Borrowing economic activity density is highly dependent on location-related factors. Areas with higher economic activity density are mostly located in the center of the study area, while those with lower economic activity density are mostly located at the edge of the study area. The number of prefecture-level cities with higher levels of borrowing economic activity density has gradually increased over time.

Second, the overall impact of borrowing population, borrowing advanced functions, and borrowing economic activity density on the economic development of small and medium-sized cities is significantly positive, indicating that borrowing size plays a positive role in the economic development of these cities. In contrast, the effect of borrowing size on large cities is mainly reflected in borrowing population, while borrowing advanced functions and borrowing economic activity density do not have significant effects on the economic development of large cities. The research provides support for the positive impact of borrowing population on the economic development of small and medium-sized cities in Hypothesis 1, but it does not prove that the effect between them is relatively insignificant. Hypothesis 2 proposes that the increase of borrowing economic activity density can promote the economic development of small and medium-sized cities, and it is confirmed in this article. Hypothesis 3 proposes that borrowing advanced functions has a dual effect on the promotion of the economic development of small and medium-sized cities. The research in this paper provides support for the positive effect, but it does not confirm the negative relationship between them.

Third, the influence of borrowing size on the economic development of small and medium-sized cities varies significantly by region. From the perspective of the four traditional major plates, for small and medium-sized cities in the central and western regions, the most positive influences are borrowing advanced functions and borrowing economic activity density, while borrowing population does not play a significant role. For small and medium-sized cities in the eastern region, the most positive influence is borrowing population, while borrowing advanced functions plays a negative role and the impact of borrowing economic activity density is not significant. For small and medium-sized cities in the eastern and northern regions, the most positive influence is borrowing economic activity density, while the impact of borrowing population and borrowing advanced functions is not significant. From the perspective of south–north differences, for small and medium-sized cities in the north, borrowing advanced functions and borrowing economic activity density have had a significantly positive impact on productivity, while borrowing population has not. However, for small and medium-sized cities in the south, the impact of borrowing population and borrowing advanced functions is highly significant, although borrowing population has had a positive effect, while borrowing advanced functions has had a negative effect. The effect of borrowing economic activity density is not significant.

### 5.2. Policy Suggestions

Based on the findings of our study, the following policy suggestions are proposed.

First, the government should continue to upgrade the road network to improve the level of accessibility among cities. The improvement of the urban network will enable borrowing size to play a positive role in urban economic development [68]. The increasingly close connections between cities has seen the level of inter-city accessibility improve, which enables small and medium-sized cities to exploit their comparative advantages. In a strong city network, these cities avoid the disadvantages of their relatively small scale by borrowing size to obtain the product markets and factor markets needed for development, and strengthen the level of exchange and cooperation with each node city in the city network. This not only enables the free flow of traditional elements such as raw materials, capital, and labor, but also promotes the exchange of knowledge, technology, and information, thereby improving the quality and efficiency of industrial development.

The distribution of urban industries should be designed to promote urban economic development. Borrowing size can play a positive role in urban economic development through benign interactions among industries in various cities. At present, it is difficult for cities to individually adapt to the development of the global economy. Thus, the development of urban groups should be seen as the future direction for the development of cities. Each city should clarify its position based on its relative strengths. For central cities, it is necessary to enhance high-end service functions, overcome difficulties in mastering core technologies, and establish an industrial structure dominated by modern service industries. For small and medium-sized cities, it is necessary to make full use of the advantages presented by low factor costs such as those related to land and labor, and to excel in attracting related industries by optimizing the business environment. Small and medium-sized cities should also cultivate traditional industries suited to local conditions and continue to consolidate the manufacturing base. Cities with an inefficient industrial layout and a lack of suitable industries should make use of urban networks to obtain access to complementary urban functions and coordinated industrial development.

Third, governments should actively cultivate and introduce talent to support the development of cities. Talent is a key factor enabling borrowing size to play a positive role in urban economic development. Talent also plays a key supporting role in relation to regional development and industrial development as an intermediary enabling small and medium-sized cities to borrow size. Generally, talented workers are, to a large extent, the carriers of knowledge and technology to a city. Through the city network, talent can take advantage of intercity exchanges to quickly access market information and advanced technologies. Through industrial cooperation, human resources can promote the inflow of

capital, which can be invested in the production of marketable products, and thus help to transform the potential market in these cities into consumer markets, thereby promoting the development of these cities. Therefore, small and medium-sized cities should implement flexible programs aimed at introducing high-level talent. These programs should take various forms such as consultant guidance, project cooperation, retirement reemployment, and talent leasing based on the requirements of local industries. They should also focus on attracting college graduates, making full use of the opportunities presented by campus recruitment programs, and undertake talent introduction activities. In addition, it is important to make suitable arrangements for the accommodation of the introduced talent to ensure that they can operate at their full capacity to promote the development of these cities.

Based on the multiple dimensions of borrowing size, this paper explores the impact of borrowing size on the economic development of small and medium-sized cities in China from the theoretical and empirical aspects. The fixed effects model of panel data is used to empirically test the impact of borrowing size on the economic development of small and medium-sized cities, enriching the quantitative empirical research on related issues. At the same time, it also provides a reference for the research of other developing countries and enriches the theory related to the impact of borrowing size on the development of small and medium-sized cities.

However, due to the availability of data and the limited level of this research, this paper still has some shortcomings: (1) In the theoretical discussion part, this article has not yet established the corresponding mathematical model from the classical theories, but only conducted some qualitative analysis at the theoretical level. (2) In terms of research perspectives, the effect of borrowing size is essentially dependent on microscopic subjects such as individuals and enterprises. However, the research in this article is not at such a micro level. (3) In terms of research scale, the major coverage of this research prefecture-level cities. Instead of using population as classification standard, this article simply classifies prefecture-level cities other than provincial capitals, municipalities, and the Shenzhen City as small and medium-sized cities. Besides, most of the small and medium-sized cities, especially small cities, exist in the form of county-level cities, and this article has not yet detailed the research to county-level cities. (4) In terms of the distance weight setting of the city network, the data used in this article are mainly the shortest road mileage available in September 2019, failing to reflect the development of the city network connection, such as the upgrade of urban roads, the process of urban road networks to be densified, and the addition of airlines and railways. Besides, the research does not include the construction and connection of inter-city communication networks.

In order to provide a more comprehensive view, future research can be conducted from the following aspects: (1) In terms of theoretical analysis, the modeling methods of classic theories can be used to construct a mathematical model of the influence of borrowing size on the economic development of small and medium-sized cities, making the argument more rigorous and powerful. (2) In terms of research perspectives, research and elaboration are suggested to be carried out from the perspective of individuals and enterprises. (3) In terms of research scale, collecting county-level data can make the research more refined. (4) In terms of setting the distance weight of the city network, it is necessary to collect more information about the construction of the transportation network between cities, and incorporate the connection of the information network between cities to construct a comprehensive and dynamic distance weight table, so as to reflect the situation of city network connection more accurately.

**Author Contributions:** Conceptualization, X.G. and F.Z.; Methodology, F.Z.; Writing—Original Draft Preparation, X.G. and F.Z.; Writing—Review and Editing, F.Z. All authors have read and agreed to the published version of the manuscript.

**Funding:** This research was supported by the Social Science Foundation of China (grant number 20XJL008).

**Data Availability Statement:** The publicly available sources for the data used in this study have been described in the article; for other data, please contact the corresponding author based on reasonable grounds.

**Acknowledgments:** Xiaojiang Ding and Peixian Liu, master students of the School of Economics, Lanzhou University, are gratefully acknowledged for their assistance in the compilation and calculation of the original data, and in the writing of the manuscript.

**Conflicts of Interest:** The authors declare no conflict of interest.

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
