# Peer review of "The Impact of Borrowing Size on the Economic Development of Small and Medium-Sized Cities in China"

_land, doi:10.3390/land10020134_

Round 1
Reviewer 1 Report
Interesting work. Generally well written and documented. However, I have three general comments to it:
1) the discussion part is missing - To what extent do the presented results confirm previous research and to what extent do they provide new information? What is the positive side of the presented research? What is their disadvantage? How do they fit into the gap marked in the abstract? What aspects should be clarified at the stage of further research?
2) At work there are places written in the personal mode (lines: 20, 565-569) in my opinion, the impersonal mode should be used.
3) Conclusions should contain direct answers to the adopted hypotheses.
Reviewer 2 Report
I must congratulate the authors for the proposal of article presented which, in general terms, I consider to be ambitious, original and innovative, but simultaneously performed in a very rigorous form. In particular, it is necessary to highlight a particularly well adapted literature review, a complex and well-presented methodology, as well as an extensive use of statistical sources in particularly extensive databases. All these are particularly praiseworthy aspects of the presented proposal.
I only have one important doubt about the text . It refers to the use of a panel data fixed effects methodology. This methodology, in general, is used for two purposes. The first is to solve the inefficiency problems presented, in these cases, by ordinary least squares estimators. The second (and, in my opinion, more important) is the estimation of unobserved heterogeneity, that is, the calculation of the observable individual effects, that is, in the case we are dealing with, in each city. The question is, if this methodology has been used, why not present the individual effects and draw the appropriate conclusions? For example, cities with more and with less productivity (individual effect), the spatial distribution of these fixed effects, etc. This is an output that is expected when it is said that a methodology based on fixed effects is going to be used. However, it does not appear in the text. Some reference is made in Table 1, but only 5 fixed effects are introduced, when we are talking about 285 cities. In this sense, data presented in table 1 rather suggest that separate regressions have been performed for groups of cities grouped according to the value of their fixed effects. But this is an assumption, because it is not clear from the text.
This would be the only major problem I see with the proposal. However, I would also make some minor suggestions that may help to refine certain aspects of the article. In particular, point 4.3. It becomes a bit tedious and difficult to read. I would recommend, a more stylized wording that would insist on the main points. Similarly, the writing of lines 301 to 304 is confusing, talking about the meaning of variables that have already been previously presented.
Otherwise, I think it is an excellent job.
Reviewer 3 Report
General comment
The paper is interesting. The authors pursue the objective of analyzing the impact of borrowing size on the economic development of small and medium-sized cities from the perspective of the national urban network. Based on the approach of Yao and Song, they defined the size of the loan as the process by which a given city accesses scale advantages by participating in a network with other cities or regions. The loan size includes three dimensions: borrowing population, borrowing economic activity density and borrowing advanced function.
They introduce the theme well with reference to literature and structure the paper in an appropriate way.
They introduce the general question and the concept of borrowing. They state the source of the data, the software used, the measurement of the borrowing size and the model to explore the three dimensions of the borrowing size, introduce the index selection and data description.
In section 4.Results they highlight the spatiotemporal evolution of the nighttime lighting density and the borrowing size, the spatiotemporal distribution of borrowing population, the spatiotemporal distribution of borrowing advanced functions, the spatiotemporal distribution of borrowing economic activity density, the results of regression analysis and provide a deepening of their by the spatial heterogeneity.
The authors comprehensively received the analysis of the city network at prefecture level in China and measured the three dimensions for borrowing size in 2004, 2009 and 2013. In particular, they proposed an interesting analysis based on socio-economic data and nighttime lighting for China’s prefecture levels cities in 2004, 2009 and 2013 and using a fixed-effect model and panel data. On the basis of these they proposed interesting comparisons between the regions of north-east, east, central, south, west and north highlight the effects of borrowing size on the economic development of the small and medium-sized cities.
On the basis of these analyses, they proposed a set of instrumental suggestions to support the choices of policy makers on the issue of territorial development, emphasizing how a better territorial accessibility can favor the coordinated development between cities with different dimensions.
Specific comment
I suggest to specify the meaning of fe1, fe2, fe3, fe4, fe5 in Table 1 and to correct the time reference for which the analysis was conducted, which covered 2004, 2009 and 2013 and not from 2004 to 2013 as noted in some parts of the text.
